# Broadband Flexible Microstrip Antenna Array with Conformal Load-Bearing Structure

**DOI:** 10.3390/mi14020403

**Published:** 2023-02-08

**Authors:** Can Tang, Hongxing Zheng, Ziwei Li, Kanglong Zhang, Mengjun Wang, Chao Fan, Erping Li

**Affiliations:** 1State Key Laboratory of Reliability and Intelligence of Electrical Equipment, Hebei University of Technology, Tianjin 300132, China; 2School of Electronics and Information Engineering, Hebei University of Technology, Tianjin 300401, China; 3School of Electronics and Information, Hangzhou Dianzi University, Hangzhou 310018, China; 4Zhejiang University-University of Illinois at Urbana-Champaign Institute, Zhejiang University, Haining 314400, China

**Keywords:** conformal load-bearing structure, microstrip antenna array, broadband, mechanical test

## Abstract

To enhance the load-bearing mechanical properties and broadband electromagnetic characteristics of the conformal antenna, a broadband microstrip antenna array with a conformal load-bearing structure is proposed in this paper, which consists of three flexible substrate layers and two honeycomb core layers stacked on each other. By combining the antenna and honeycomb core layer in a structural perspective, the antenna array is implemented in the composition function of surface conformability and load-bearing. Additionally, the sidelobe level of the antenna is suppressed based on the reflection surface loaded. Meanwhile, an equivalent model of a honeycomb core layer has been established and applied in the design of a conformal antenna with a load-bearing structure. The presented model increases the accuracy of simulated results and reduces the memory consumption and time of the simulation. The overall size of the proposed antenna array is 32.84 × 36.65 × 4.9 mm (1.36 λ_0_ × 1.52 λ_0_ × 0.2 λ_0_, λ_0_ is the wavelength at 12.5 GHz). The proposed antenna element and array have been fabricated and measured in the flat state and under other various bending states. Experiment results show the operating relative bandwidth of the antenna array is 20.68% (11.67–13.76 GHz and 14.33–14,83 GHz) in the flat state. Under different bending conditions, the proposed antenna array covers 24.16% (11.08–14.1 GHz), 23.82% (10.63–13.5 GHz), and 23.12% with 30°, 60°, and 90° in the *xoz* plane (11.55–14.33 GHz). In terms of mechanical load bearing, the structure has better performance than the traditional single-layer honeycomb core load-bearing structure antenna.

## 1. Introduction

With the benefit of compatibility with the surface of carriers, the conformal antenna can ensure the electromagnetic characteristics of the antenna while effectively meeting the good aerodynamic layout of the carriers [1]. Achieving electromagnetic (EM) radiation characteristics without destroying the original aerodynamic layout of the carrier, the conformal antenna has been used extensively in airplanes, tanks, and various complicated and composite platforms [2]. Due to the increased complexity of the carriers and the diversity of load forms, the requirements for the conformal antenna are different. There are two significant components in the conformal load-bearing antenna (CLA): flexible antennas and a load-bearing structure. On the one hand, the conformal antenna is embedded under the skin of the aircraft with high integration, which demands the design of antennas with a miniaturized structure, stable radiation characteristics, and the capacity to bear dynamic loads [3]. On the other hand, taking manufacturing cost, portability, and life cycle into consideration, the design of a conformal antenna should be simple in structure and flexible in load-bearing [4]. Kim et al. [5] investigated the impact response and damage of the conformal antenna with a load-bearing structure, which indicated the relationship between impact damage and antenna performance, and the impact energy thresholds for acceptable antenna functions were 1.5 and 1.75J. Baek et al. [6] proposed a conformal load-bearing structure for an antenna array and pointed out that a multi-layered structure could be a good candidate for the CLA design. Using a multilayer structure, Bishop et al. [7] provided a broadband high-gain bi-layer antenna for CLA application without specific research on mechanical properties. In fact, ongoing research mainly focuses on the design of conformal antenna using flexible material, and the load-bearing structure is ignored [5,6,7].

A CLA is developed at the system level, rather than a single design of the antenna [8,9]. The honeycomb core layer is often applied in the design of the load-bearing structure due to its light weight and great mechanical load-bearing properties. A method for measuring the mechanical properties of the antenna is presented by Healey et al. [9], including tensile, biaxial, and twisting. Nevertheless, experimental verification is lacking. Meanwhile, the conformal load-bearing antenna structure for mechanical loading consideration has been reviewed in this paper. Wang et al. [10] proposed a novel e-fiber antenna embedded in a polymer substrate for load-bearing. The conventional conformal antenna with load-bearing function has been depicted in Figure 1. It is composed of four parts: flexible antennas, honeycomb core layer, upper face sheet, and lower face sheet. Firstly, flexible antennas are in the middle layer as electronic devices. Initially filled between the face sheet and antennas, honeycomb core layers play a crucial role in the load-bearing function. Finally, the upper face sheet and lower face sheet are added as the surface of the conformal load-bearing structure [3,10]. 

With the advantages of a lightweight, low-profile, and simple structure, the flexible microstrip antenna array has been broadly applied in the design of various conformal antenna. The narrow impedance bandwidth is the main limitation of the design of conformal antenna [11,12,13]. Based on the microstrip antenna, several related technical methods for bandwidth enhancement have been studied and verified. Owing to the diamond-shaped slot, Huynh et al. [14] proposed a broadband microstrip antenna with a fractional bandwidth of 13.6%. Through the planar multilayer configuration, Sun et al. [15] provide a feasible solution to improve the bandwidth of the microstrip antenna. However, these methods were employed for rigid materials [14,15,16,17]. In terms of theory, the bandwidth of the microstrip antenna can be broadened by increasing its profile and reducing the dielectric constant (*ε_r_*) of the substrate [18]. Meanwhile, the reliability and stability of the antenna are important indicators of its application prospects. Hence, the simple design of the antenna structure can effectively improve its stability and reliability [1].

In this paper, an efficient and simplified method to realize CLAs with broadband has been presented. The proposed antenna array consists of a Wilkinson power divider circuit and four microstrip antenna elements which are composed of three layers of the flexible substrate and two layers of the honeycomb core. Stacked between three flexible substrates, the honeycomb layer and antenna are integrated into the structure, which broadens the bandwidth of the antenna in terms of structure. Then, the principle of its wideband operating is analyzed through the characteristic mode analysis (CMA) theory. Meanwhile, the sidelobe level (SLL) is reduced by reflection metal printed on the back of the lower substrate. Additionally, it is known that the EM parameters of the honeycomb core layer have a great influence on the performance of the conformal antenna [19,20,21,22]. For this purpose, an equivalent medium model of the honeycomb core layer is proposed for the simulation of the conformal load-bearing structure. Furthermore, the mechanical properties of the proposed structure have been verified through the three-point bending test and the quasi-static compression test. In Section 2, the antenna element and array design are described. Section 3 introduces the equivalent medium model of the honeycomb core layer and the principle for wideband operation. Section 4 illustrates the simulated and measured results of the analysis of fabricated antenna, including the radiation characteristics and mechanical performance, followed by the comparison. The work has been concluded in Section 5.

## 2. Antenna Design

### 2.1. Antenna Element

The overall geometry of the proposed antenna element is shown in Figure 2. Composed of three substrate layers (polyimide film, *ε_r_* = 3.5, tanδ = 0.0027, thickness = 0.3 mm), four metallic layers, and two honeycomb core layers (*ε_r_* = 1.1, thickness = 2 mm), the layered structure was depicted Figure 1 in perspective view. Specifically, the relative positions of these layers are as follows: (1) radiation patch on the top surface of the upper substrate layer; (2) ground plane with coupling aperture and microstrip line on the top and lower surface of the middle layer substrate, respectively; (3) metallic reflection surface on the back surface of the lower surface; and (4) honeycomb cores between the three substrate layers. The radiation patch is excited by a 50 Ω microstrip feeding through a coupling aperture and the metallic reflection surface is used to reduce the SLL of the antenna element.

Figure 2b,c shows the structure of the honeycomb core and the side view of the antenna element, respectively. The mixed equivalent permittivity of the upper layer of substrate (*ε_r_*) can be expressed as:(1)$$\varepsilon_{r}=\frac{h_{1}+h_{2}}{\frac{h_{1}}{\varepsilon_{1}}+\frac{h_{2}}{\varepsilon_{2}}}$$

Then, *H_p_* is the mixed thickness of layers, the relationships between *BW*, *H_p_*, and *ε_r_* are described as [18]:(2)$$H_{p}=h_{1}+h_{2},$$
(3)$$BW\propto \left(1/\varepsilon_{r}\right);\;BW\propto H_{p},$$
where *h*_1_ and *h*_2_ are the thickness of substrate layers and honeycomb core layers, *ε*_1_ and *ε*_2_ are the equivalent permittivity of the substrate layer and honeycomb core layer. The *BW* of the microstrip antenna is proportional to *H_p_* and is inversely proportional to *ε_r_*. Therefore, loading honeycomb layers plays two vital roles in the proposed element with a conformal load-bearing structure. For one thing, it takes the load-bearing function and stability for the antenna. For another, it broadens the BW through the increase in *H_p_* and the decrease in *ε_r_*. Calculated by Formulaes (1) and (3), the bandwidth of the proposed antenna element is increased by the loading of the honeycomb core between the upper and middle substrate layer.

From Figure 2d, the length and width of the radiation patch is *L*_2_ × *W*_2,_ which can be calculated by [18]:(4)$$L_{2}=\frac{C}{2f_{r}\sqrt{\varepsilon_{e}}}-2\Delta L_{p},\;W_{2}=\frac{C}{2f_{r}}\sqrt{\frac{2}{\varepsilon_{r}+1}},$$
(5)$$\varepsilon_{e}=\frac{\varepsilon_{r}+1}{2}+\frac{\varepsilon_{r}-1}{2}\frac{1}{\sqrt{\left(1+12H_{p}/W\right)}},$$
(6)$$\Delta L_{p}=0.412H_{p}\frac{\varepsilon_{e}+0.3}{\varepsilon_{e}-0.258}\frac{\frac{W_{P}}{H_{P}}+0.264}{\frac{W_{P}}{H_{P}}+0.800},$$
where *f_r_*, *C*, Δ*L_p_*, *W_p_*, *ε_e_*, and *ε_r_* are the operating frequency point of the antenna, the speed of light, the amount of extension of the patch, the width of the 50 Ω microstrip feeding line the mixed relative effective permittivity, and the mixed equivalent permittivity, respectively. 

In Figure 2e, the coupling aperture is implemented by etching a rectangular slot in the center of the ground plane, and its dimension is *L*_3_ × *W*_3_. The dimensions of the 50 Ω microstrip feeding line are *L*_4_ × *W*_4_ which is printed on the lower surface of the middle substrate. Additionally, the overall size of three substrate layers is *L*_1_× *W*_1_ × *h*_1_, and the overall dimension of two honeycomb cone layers is *L*_1_ × *W*_1_ × *h*_2_. 

### 2.2. Antenna Array

As shown in Figure 3, the composition of the antenna array includes four elements designed above, and a four-way Wilkinson power divider with equal amplitude and phase. The proposed four antenna elements are configured with 2 × 2 in equal spacing *d*_1_, and the four-way Wilkinson power divider is printed on the lower surface of the middle substrate, connecting with the microstrip feedings of the proposed four antenna elements. Due to the independence of the feeding network, the Wilkinson power divider can be designed separately, which makes the antenna array easy to extended. 

## 3. Theoretical Analysis

### 3.1. The Equivalent Medium Model of the Honeycomb Core Layer

The typical structure of the honeycomb core layer with a periodic cell and three-dimensional (3D) geometry has been shown in Figure 4. The periodic cell is a regular hexagon, where *C*_1_ and *h*_2_ are applied to describe the side length and thickness of the ortho-hexagonal cell, and *t* is the thickness of the cell wall, respectively. The honeycomb layer consists of the host material (Nomex, *ε_r_* = 4) and filling material (air, *ε_r_* = 1) in terms of material.

The honeycomb core layer is an EM transmission device. Consequently, we mainly focus on its radial-effective permittivity and permeability in this work. The solution physical model and the process of the solution method are shown in Figure 5. In the solution physical model, it should be noted that port 1 and port 2 are immediately below and above the honeycomb core layer, respectively. Then, in process of the presented solution method, the equivalent EM parameters of the homogeneous medium plane can be obtained from the scattering parameters of the two-port. Illuminated by an incident plane wave, the layer of a homogeneous material with the characteristic impedance *Z*, and the refractive index *n* can be expressed as: (7)$$Z=\sqrt{\frac{{\left(1+S_{11}\right)}^{2}-S_{21}^{2}}{{\left(1-S_{11}\right)}^{2}-S_{21}^{2}}}$$
(8)$$n=\frac{1}{kh_{2}}\cos^{-1}\left[\frac{1}{2S_{21}}\left(1-S_{11}^{2}+S_{21}^{2}\right)\right]$$
where *h*_2_ is the thickness of the honeycomb layer, and *k* denotes the wave number in free space. The effective permittivity *ε_e_* and permeability *μ_e_* of the planar homogeneous material is given by:(9)$$\varepsilon_{e}=\frac{n}{Z}$$
(10)$$\mu_{e}=n\cdot Z$$

According to a two-port scattering parameter retrieval method (Nicolson–Ross–Weir) [23], the honeycomb periodic cell structure and layer are established as physical models for the simulation first. In these models, the honeycomb layer is composed of 5 × 10 periodic cells, and the dimension of the cell structure is as follows: *C*_1_ = 1.83 mm, *h*_2_ = 2 mm, and *t* = 0.035 mm. Following this, these models are both simulated by CST MWS at a frequency band from 10 GHz to 15 GHz. The results of the honeycomb periodic cell and layer, including effective permittivity in the real part (*ε_e_*), effective permittivity in the imaginary part (*ε_e_*′), effective permeability in the real part (*μ_e_*), and effective permeability in the imaginary part (*μ_e_*′), are depicted in Figure 6. It can be seen that there are some slight differences between the periodic cell and layer in the calculated effective permittivity and permeability. 

To compare the validity and reasonableness of the EM parameters in the equivalent homogeneous medium plane, the reflection and transmission coefficients of these equivalent plane and honeycomb core layer structures are simulated in flat state and under different bending states. The results of the comparison have been shown in Figure 7. The maximum errors in the reflection coefficient of the cell equation and layer equation are within 3 dB and 1 dB, respectively. In the transmission coefficient and the maximum error of the cell equation and layer equation are within 0.01 dB and 0.004 dB, respectively. Thus, the layer equivalent is in good agreement with the honeycomb core layer in the EM performances.

In short, the equivalent EM parameters of the honeycomb core layer can be effectively extracted in the frequency domain through full-wave analysis. Then, the axial homogeneous layer can be established through the equivalent EM parameters. Hence, the equivalent homogeneous medium plane can be used for the design and simulation of CLAS due to its high accuracy in EM performance, simplicity in the model, and simulation efficiency. 

### 3.2. Principle for Wideband Operation

In order to better reveal the broadband operation mechanism of the antenna, the characteristic mode analysis (CMA) theory of the antenna structure is carried out using a CST microwave studio (version 2016). The CMA theory was proposed by Garbacz in the late 1960s, who suggested that the eigen currents could be determined by analyzing the scattering matrix. These characteristic currents are surface currents on the conducting body and depend only on the shape of the conductor without any external excitation. The model significance indicates the normalized magnitude of the eigen currents and the band with MS greater than 0.707 is the potential operating band of the antenna. Thus, the CMA analysis allows the analysis of the potential operating bandwidth of the antenna structure.

Firstly, the CMA on the radiation patch and the top substrate is carried out, as shown in Figure 8. For these modes, only mode 1 and mode 2 have identical modal significance. It can be observed from Figure 8b,c that the modal currents of mode 1 and mode 2 show orthogonal characteristics. Due to the difference in the width and length of the radiation patch, the difference occurs in the resonant frequency points of mode 1 and mode 2. As illustrated in Figure 8a, the operating frequency band with an MS greater than 0.707 is extremely narrow in both modes, and this structure for the antenna design is a hard strategy to implement a broadband operating frequency band. Initially, the CMA method is employed after the honeycomb core equivalent layer is added, as depicted in Figure 9. It is apparent that mode 1 is identical to mode 2 in the modal significance. In this case, the potential operating frequency band is wider than the previous structure which is shown in Figure 8a. Meanwhile, these results further support the concept in formulae (3) that the operating bandwidth of the microstrip antenna can be improved by adding a substrate layer with a low dielectric constant between the radiation patch and the ground.

Finally, to further investigate the operating bandwidth stability under deformation, its structure utilizes the CMA method to analyze various bending states as shown in Figure 10a. Since prior cases indicate that mode 1 and mode 2 are degenerate modes, only one of them is needed for the analysis. In this case, mode tracing is performed for mode 1, and the modal significance has been shown in Figure 10b. Whether in a flat or deformed state, this structure maintains a wide operating frequency band. Although the deformable state will cause a partial shift in frequency, the primary operating frequency band is out of change. 

## 4. Analysis of Simulation and Measurement Results

As an important frequency band for space satellite communications, Ku-band was used as the operating frequency band for the proposed CLA. Its central frequency point was 12.5 GHz. To demonstrate the feasibility of the design above, the proposed conformal antenna element and antenna array were simulated based on the optimized dimensions by CST, and its optimized dimensions are tabulated in Table 1. The equivalent medium model of the honeycomb core layer was applied to the simulation of the proposed antennas. Then, the prototypes were processed and measured by a vector network analyzer (PNA-X5244A) and microwave anechoic chamber in our laboratory. The proposed antenna element and antenna array were mounted on the mold through nylon screws, respectively, for conformability testing. 

### 4.1. Antenna Element

The structure photographs of the fabricated antenna element are presented in Figure 11. Observed in Figure 12, the measured reflection coefficient results of the proposed antenna element are 11.56–13.95 GHz, 11.53–14.04 GHz, 11.58–13.94 GHz, and 11.60–14.01 GHz in flat and various bending with 30°, 60°, and 90° in the *xoz* plane, respectively. As can be seen in Figure 13, the SLL is reduced by 13.4 dB in E-plane and 13.6 dB in H-plane under the reflection surface loaded, respectively. Figure 14 displays the simulated and measured gain radiation patterns and cross-polarization radiation patterns of the antenna element at 12.5 GHz. Whether in a flat or under a different bending state, the radiation direction of the proposed antenna element can be maintained as stable with low cross-polarization and their maximum radiation directions are kept in the direction perpendicular to the horizontal plane of the antenna. Its performance characteristics in flat state and under different states are shown in Table 2.

### 4.2. Antenna Array

The measurement environment and prototype of the fabricated antenna array are depicted in Figure 15. Its simulated and measured reflection coefficient has been plotted in Figure 16. On account of the mutual coupling between the antenna elements, the bandwidth of the antenna array has been broadened. In the flat state, its measured operating frequency band is 11.67–13.76 GHz and 14.33–14.825 GHz with a relative bandwidth up to 20.68%. In a bending state, the deformation of the power divider leads to impedance discontinuity in the microstrip line, and this is typically a complex problem that affects the transmission of electrical signals. Since electrical signals can be maintained by the coupled aperture, in the deformation with 30°, 60°, and 90° in the *yoz* plane, the operating frequency band of the antenna array still covers 11.08–14.1 GHz, 10.63–13.5 GHz, and 11.55–14.33 GHz, and their relative bandwidths are 24.16%, 23.82%, and 23.12%, respectively. In Figure 17, its simulated and measured gain and cross-polarization radiation patterns are shown at 12.5 GHz. With the increment of bending angle, the cross-polarization of the antenna array is boosted, and the distortion of the antenna array radiation pattern occurs. The peak gain of the antenna array keeps stable radiation at the *z*-axis, both in a flat state and under different bending states. The performance characteristics of the proposed antenna array are listed in Table 3.

### 4.3. Mechanical Properties

In order to validate the mechanical properties of the designed antenna, three-point bending and the mechanical experiment approach were employed in the following structure, including the single-layer honeycomb sandwich panel and double-layer honeycomb sandwich panel, respectively. Meanwhile, the quasi-static compression of the proposed antenna array structure has been simulated and measured. In the experiment, in order to avoid the concentration of stress, the multilayer structure composed of the substrate and the honeycomb is adhered by glue.

Figure 18 is the three-point bending experiment and simulation of the structure with a single honeycomb core layer. According to the three-point bending deformation of the single-layer honeycomb sandwich panel in Figure 18a,b, it can be found that the honeycomb core layer and substate layers have obvious deformation, and the upper and lower substrates and honeycomb core layer are not unglued. Figure 18c shows the force-displacement curve during the three-point bending of the single-layer honeycomb sandwich panel. During the bending process, there is a sudden drop in the acting force at approximately 1.3 mm, and then it becomes smooth. The force acting in bending 1.3mm is approximately 140 N. Moreover, the three-point bending test of the double-layer honeycomb sandwich panel is shown in Figure 19. Based on the deformation of Figure 19a,b, it can be noted that both honeycomb core layers and substrate layers are in line with obvious deformation, and the middle substrate layer shows a debonding phenomenon with the lower honeycomb core layer. In addition, Figure 19c details that the force of bending displacement of 1 mm is 263 N. Comparison of the results with a single-layer honeycomb sandwich panel confirms that the proposed antenna array structure is better for load bearing and bending resistance. Furthermore, the quasi-static compression test of the proposed antenna structure has been shown in Figure 20. Observed from Figure 20a, the deformation of the double-layer honeycomb panel is mainly the deformation of the honeycomb core layer. Moreover, the honeycomb deformation of the upper layer is more obvious than that of the lower layer. Figure 20c indicates that the maximum pressure that the structure can bear in the elastic deformation stage is approximately 1800 N.

In general, the simulated and measured results of the proposed antenna element and antenna array are in great agreement with each other. At the same time, the validity of the equivalent model has also been verified in measured results. The electrical performance of the antenna element and antenna array are stable whether in flat state or in deformation. Owing to the loading of physical deformation molds, the accuracy of simulation, and the measurement environment, slight deviations occur. Compared with the similar antennas in Table 4, the proposed antenna array fulfills wider bandwidth and a lower SLL than others. It can be applied in the complicated and composite platform with potential due to its simplicity in structure, light weight, and integration with functions.

## 5. Conclusions

To implement the conformal load-bearing mechanical properties and broadband radiation characteristics, a novel structure for microstrip antenna design has been proposed in this paper. Firstly, our methods shed light on a CLA with integrated functions in terms of structure. Through the addition of honeycomb core layers and metallic reflection surfaces, the proposed antenna achieves an increase in bandwidth and a decrease in the SLL. With a load-bearing function and simple structure, the proposed antenna remains stable in EM performance. Initially, the antenna array is convenient to achieve and is extended by redesigning the antenna feeding network in the middle substrate layer. Meanwhile, providing an efficient and accurate solution for the design and simulation of CLA, an equivalent model of the honeycomb core layer is presented. The consistency of simulation and measurement results certified its accuracy. Since the consideration and simulation of the practical and challenging scattering problem, the reliability and reasonableness of the CLA design have been improved efficiently. 

## Figures and Tables

**Figure 1 micromachines-14-00403-f001:**
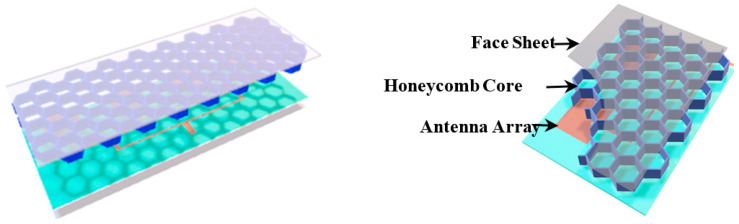
The conventional structure of conformal load-bearing antenna array.

**Figure 2 micromachines-14-00403-f002:**
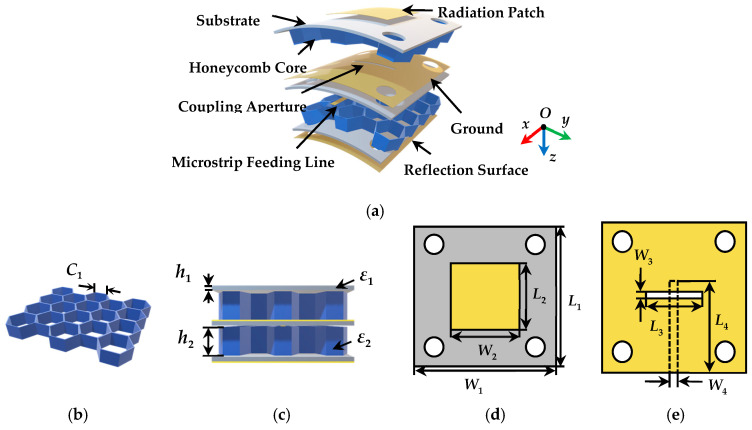
The coordinate geometry and dimensions of the proposed antenna element: (**a**) explosion view of the layered structure, (**b**) honeycomb core structure, (**c**) side view of element, (**d**) top layer with radiation patch, and (**e**) middle layer with the microstrip feeding and ground with coupling aperture.

**Figure 3 micromachines-14-00403-f003:**
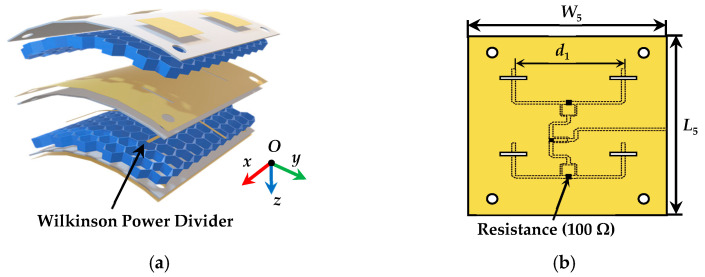
The coordinate geometry and dimensions of the proposed antenna array: (**a**) explosion view of the structure, (**b**) the feeding network with Wilkinson power divider and ground with four coupling apertures on the middle layer.

**Figure 4 micromachines-14-00403-f004:**
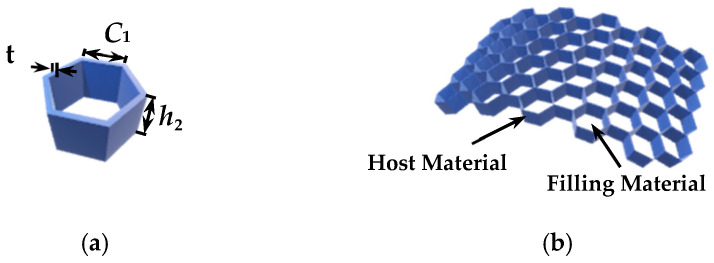
The typical structure of a honeycomb core layer: (**a**) cellular structure and (**b**) 3D conformed view.

**Figure 5 micromachines-14-00403-f005:**
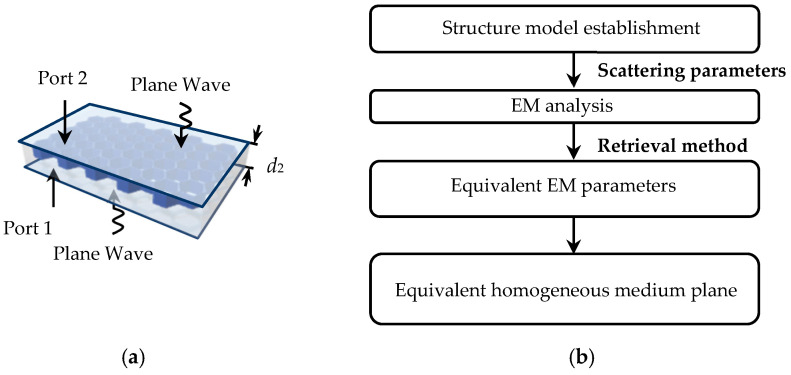
The solution process of the equivalent model, (**a**) the solution physical model, and (**b**) the process of the presented solution method.

**Figure 6 micromachines-14-00403-f006:**
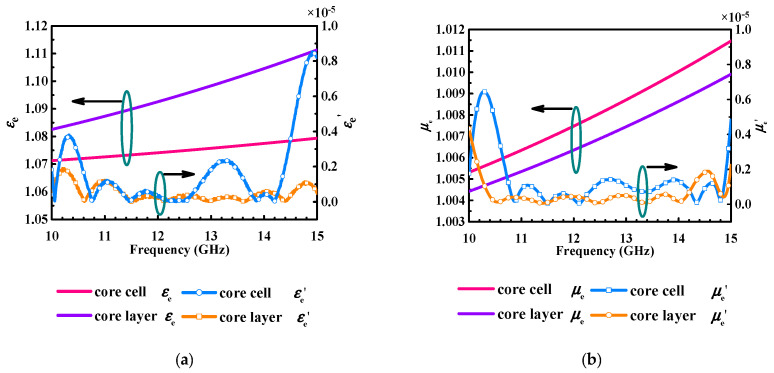
The EM parameters of periodic cell and layer in real and imaginary parts, (**a**) effective permittivity (**b**) effective permeability.

**Figure 7 micromachines-14-00403-f007:**
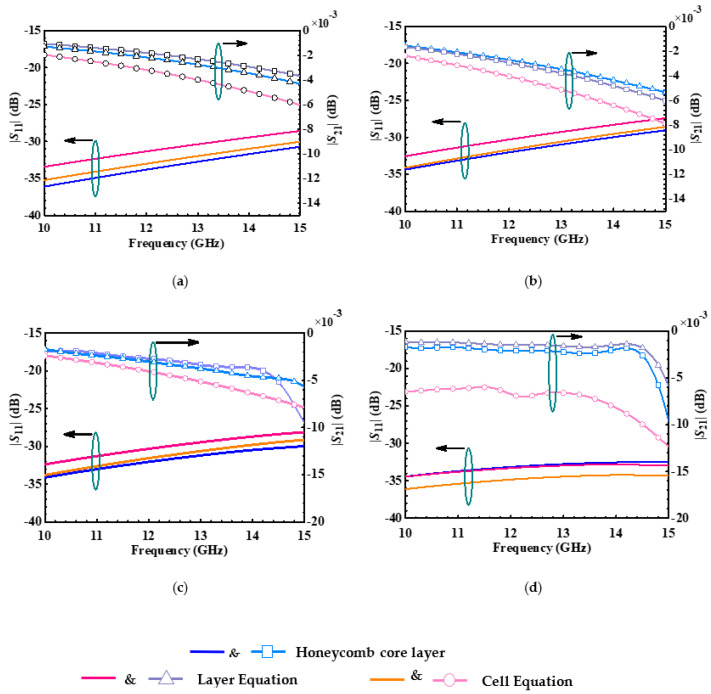
The comparison of scattering parameters of the honeycomb core layer, cell EM equation, and layer EM equation in flat state and under different states, (**a**) planar, (**b**) 30° bending, (**c**) 60° bending, and (**d**) 90° bending.

**Figure 8 micromachines-14-00403-f008:**
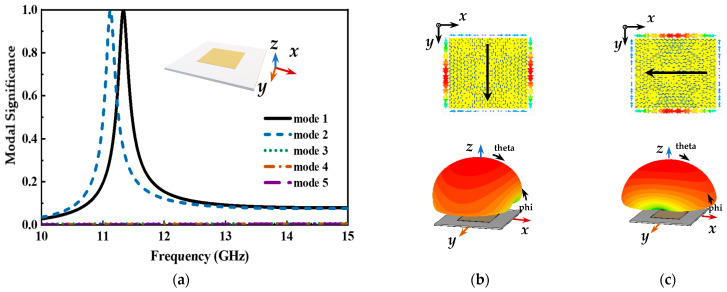
Modal behaviors of the top substrate and radiation patch, (**a**) modal significance, (**b**) the modal current and far-field radiation pattern of mode 1, and (**c**) the modal current and far-field radiation pattern of mode 2. Modal Significance.

**Figure 9 micromachines-14-00403-f009:**
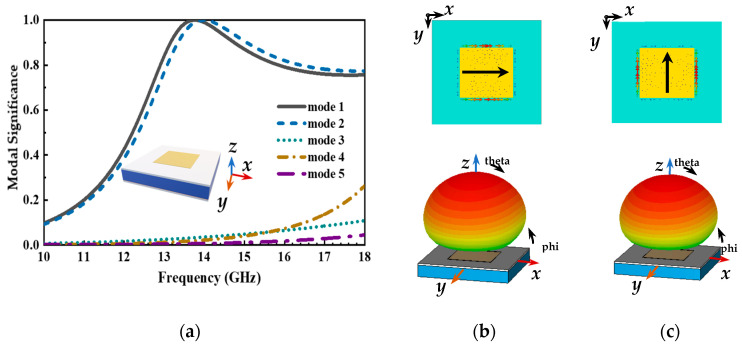
Modal behaviors of the top substrate, radiation patch, and honeycomb equivalent layer, (**a**) modal significance, (**b**) the modal current and far-field radiation pattern of mode 1, and (**c**) the modal current and far-field radiation pattern of mode 2.

**Figure 10 micromachines-14-00403-f010:**
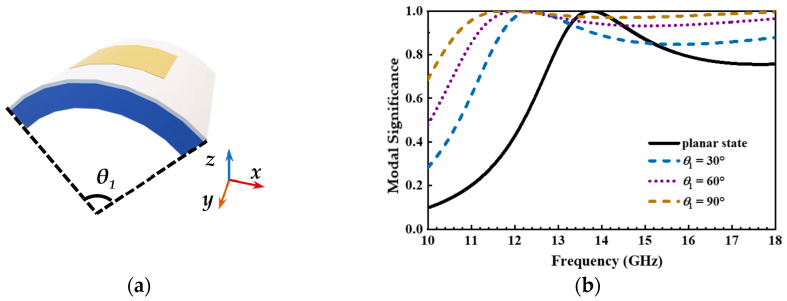
Various bending modal behaviors of the top substrate, radiation patch, and honeycomb equivalent layer, (**a**) bending states, and (**b**) modal significance.

**Figure 11 micromachines-14-00403-f011:**
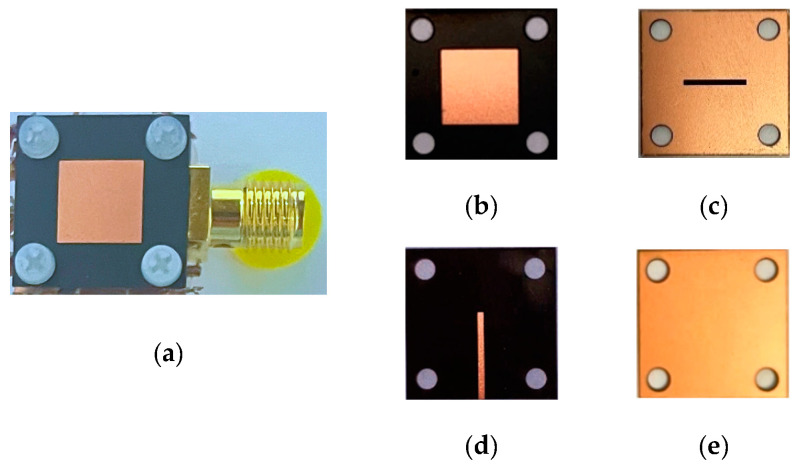
The photograph of the proposed antenna element: (**a**) top view, (**b**) top layer with radiation patch, (**c**) the ground with coupling aperture on top of the middle substrate layer, (**d**) microstrip feeding line on the back of middle layer, and (**e**) metallic reflection surface on the back surface of the lower substrate layer.

**Figure 12 micromachines-14-00403-f012:**
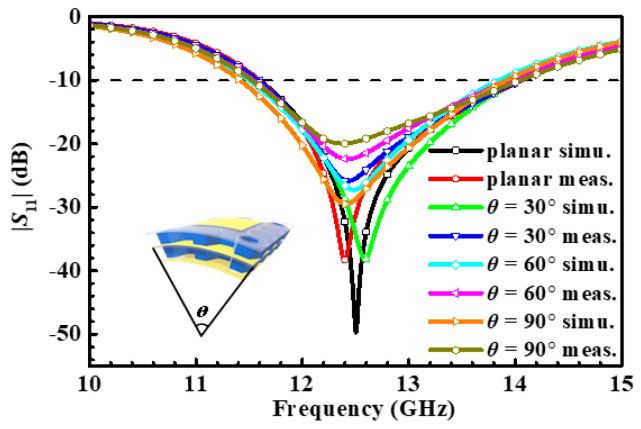
Simulated and measured reflection coefficient results of proposed antenna element under various bending states and on the plane. The inset illustrated the bending state.

**Figure 13 micromachines-14-00403-f013:**
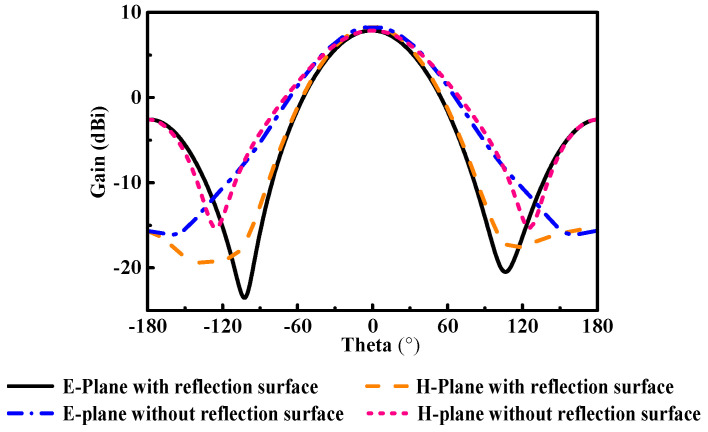
The simulated radiation pattern of antenna elements with the comparison of the reflection surface loaded both before and after.

**Figure 14 micromachines-14-00403-f014:**
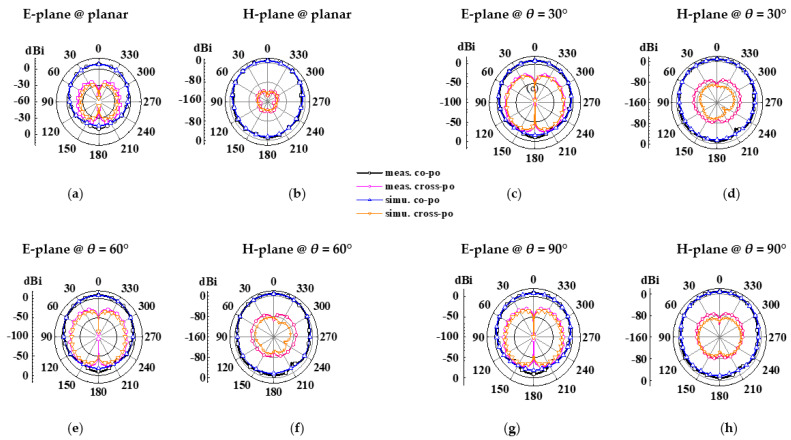
Simulated and measured results for gain radiation patterns and cross-polarization radiation patterns of the proposed antenna element in flat state and under various bending states at 12.5 GHz, (**a**–**h**) E-plane and H-plane with various states.

**Figure 15 micromachines-14-00403-f015:**
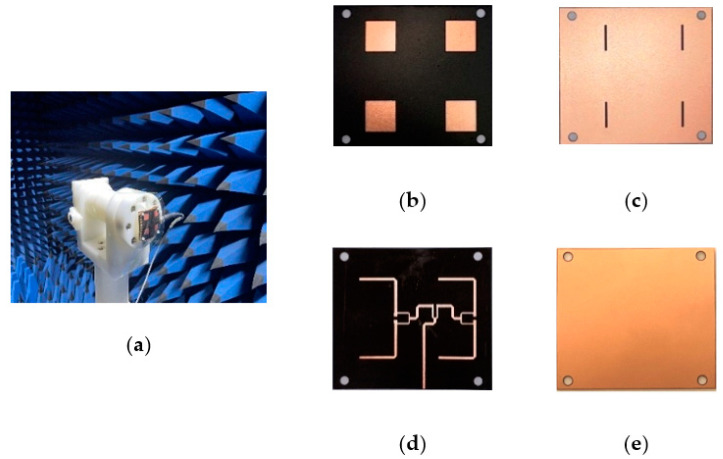
The photograph of the proposed antenna array: (**a**) measurement environment, (**b**) top layer with radiation patches, (**c**) the ground with coupling apertures on the middle layer, (**d**) the feeding network with Wilkinson power divider and ground with four coupling apertures on the middle layer, and (**e**) reflection surface on the back surface of the lower substrate layer.

**Figure 16 micromachines-14-00403-f016:**
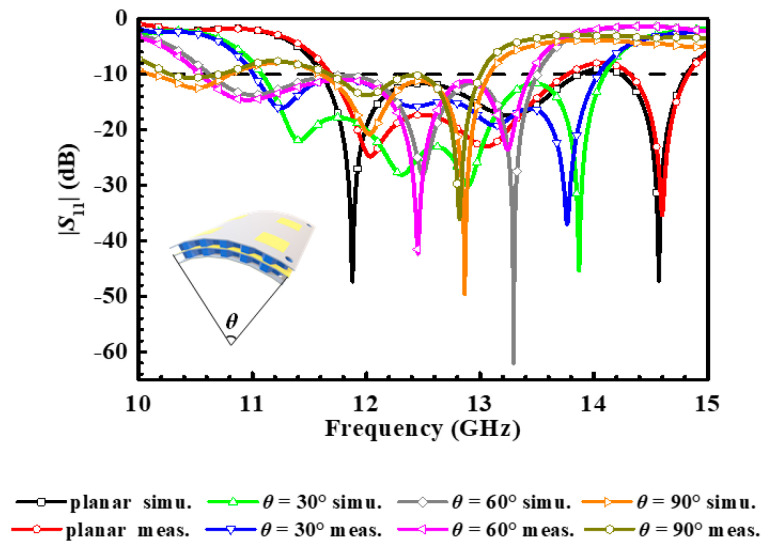
The simulated and measured reflection coefficient of the proposed antenna array under various bending states and on the plane. The inset illustrated the bending state.

**Figure 17 micromachines-14-00403-f017:**
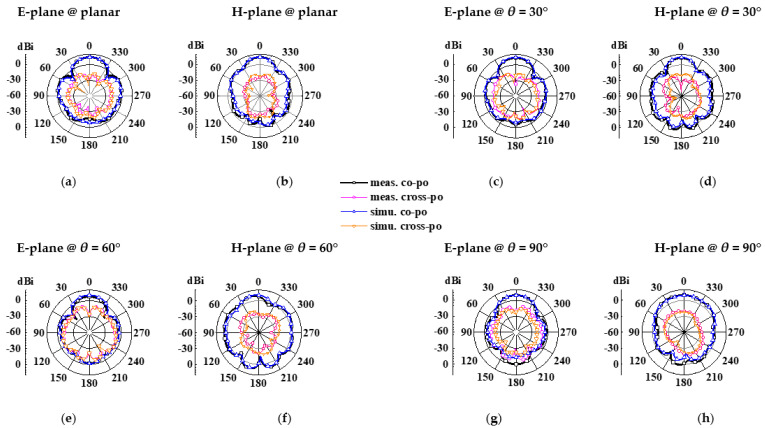
Simulated and measured results for gain radiation patterns and cross-polarization radiation patterns of the proposed antenna array in flat state and under various bending states at 12.5 GHz(**a**–**h**) E-plane and H-plane with various states.

**Figure 18 micromachines-14-00403-f018:**
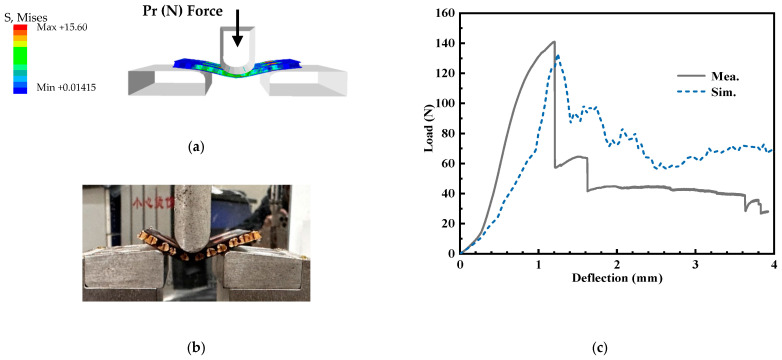
The three-point bending test of single-layer honeycomb sandwich panel, (**a**) simulation model (**b**) experiment environment, and (**c**) load–deflection relationships.

**Figure 19 micromachines-14-00403-f019:**
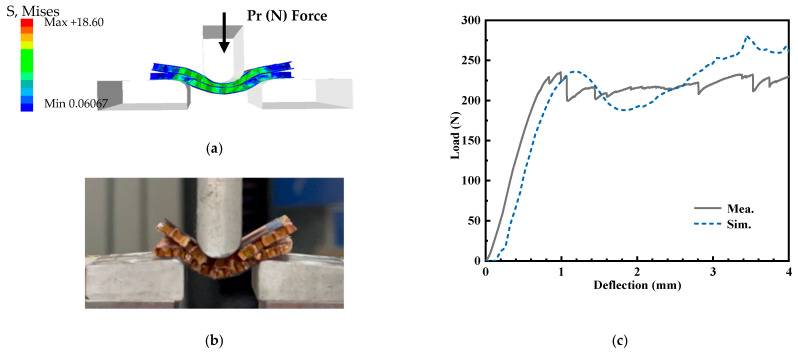
The three-point bending test of double-layer honeycomb sandwich panel, (**a**) simulation model (**b**) experiment environment, and (**c**) load–deflection relationships.

**Figure 20 micromachines-14-00403-f020:**
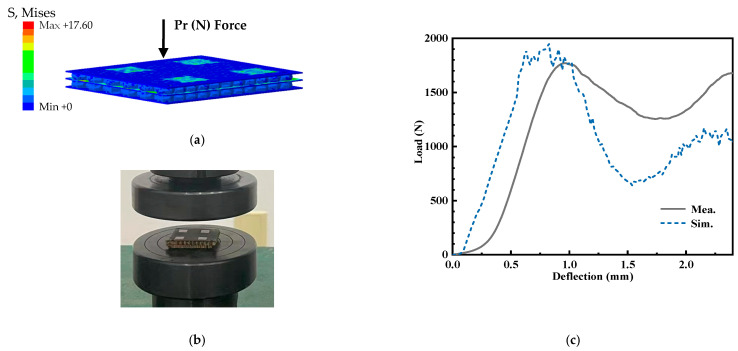
The quasi-static compression test of double-layer honeycomb sandwich panel, (**a**) simulation model (**b**) experiment environment, and (**c**) load–deflection relationships.

**Table 1 micromachines-14-00403-t001:** Optimized dimensions of CLAS antenna (unit: mm).

*W* _1_	*W* _2_	*W* _3_	*W* _4_	*W* _5_	*L* _1_	*L* _2_	*L* _3_
14.52	7.04	0.51	0.64	32.84	14.52	6.9	6.51
** *L* _4_ **	** *L* _5_ **	** *C* _1_ **	** *h* _1_ **	** *h* _2_ **	** *ε* _1_ **	** *ε* _2_ **	** *d* _1_ **
8.4	36.65	1.83	0.3	2	3.5	4	18.32

**Table 2 micromachines-14-00403-t002:** Performance characteristics of the proposed antenna element.

State	BW	Peak Gain (dBi)	3 dB Beam Angle (°)
Planar	20.68%	8.32	67.0
30°	24.16%	8.25	67.4
60°	23.82%	8.12	67.8
90°	23.12%	8.03	68.5

**Table 3 micromachines-14-00403-t003:** Performance characteristics of the proposed antenna array.

State	BW	Peak Gain (dBi)	3 dB Beam Angle (°)
Planar	20.68%	14.7	38.6
30°	24.16%	14.4	37.8
60°	23.82%	12.9	38.7
90°	23.12%	12.7	39.1

**Table 4 micromachines-14-00403-t004:** Comparison between proposed and similar conformal antenna array.

Antenna	Size (mm^3^)	Frequency (GHz)	Methods	BW	SLL (dB)	Flexible	With CLAS
Ref. [24]	46.36 × 46.36 × 1.78 (4 elements)	7.7–8.3	Inkjet printing	12.5%	−14	Yes	Yes
Ref. [25]	33 × 20 × 5 (1 element)	6.8–8.3	DRA	18.4%	−10	No	No
Ref. [26]	44 × 85 × 4.9 (4 elements)	12.2–13.4	3D printing	11.6%	−8	Yes	No
Ref. [27]	45 ×38 × 0.2 (4 elements)	2.37–2.61	Fractal structure	9.63%	-	Yes	No
		3.30–4.41		28.79%			
		4.98–5.90		16.91%			
Ref. [28]	59 × 29 × 0.1 (2 elements)	2.38–2.55	Inverted-F branch	7.2%	-	Yes	No
		3.37–3.60		6.4%			
		4.92–5.37		9.3%			
Ref. [29]	32 × 40 × 4.9 (4 elements)	2.36–2.87	Fractal structure and multi-branch	15.68%	-	Yes	No
		3.4–12		111.69%			
Ref. [30]	55 × 55 × 0.2 (1 element)	3.34–5.01	Multi-branch	40%	-	Yes	No
		8.9–9.2		3.31%			
**This work**	**32.84 × 36.65 × 4.9** **(4 elements)**	**11.67–13.76 and 14.33–14.83** **(planar state)**	**Aperture-coupled feeding**	**20.68%**	**−24**	**Yes**	**Yes**
		**11.08–14.1 (30° bending)**		**24.16%**			
		**10.63–13.5 (60° bending)**		**23.82%**			
		**1.55–14.33 (90° bending)**		**23.12%**			

## Data Availability

The data supporting the findings of this study can be made available to the genuine readers after contacting the corresponding authors.

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
