# Peer review of "Broadband Flexible Microstrip Antenna Array with Conformal Load-Bearing Structure"

_micromachines, 2023, doi:10.3390/mi14020403_

Round 1

Reviewer 1 Report

This paper presents a design of a flexible broadband microstrip antenna array. Also, it presents a detailed analysis of conformal load-bearing structure which is supported by experimental results. This paper is actually of sufficient interest because of its comprehensive analysis. However, there are some recommendations suggested to improve the quality of the proposed work.

Minor Corrections:

1.     Line no. 59: structures --> structure

2.     Line no. 75: an feasible --> a feasible

3.     Line no. 136: Light speed is represented by upper and lower case letters C and c (Ref Eq 4).

4.     Line No. 148: "four antenna elements are configurated with 2×2 in equal spacing d1"

Configurated --> configured

d1--> (as per Figure 3b)

5.     Line 150: “Due to the independence of the feeding network, the Wilkinson power divider can be designed separately, which makes the antenna array is easy to extend.” Re-phase the sentence.

6.     Line no. 160 (184): “where a and hc are applied to describe the side length and thickness of the ortho-hexagonal”. In Figure 2 these parameters are described as C1 and h2. Please do the needful corrections.

7.     Line no. 178: “d is the thickness of the honeycomb layer”. The honeycomb thickness is denoted by h2, whereas the d is the antenna array elements spacing. Kindly change the formula notations as per the considered notations.

8.     Line no. 243: virous --> various

9.     Line no. 298: the bending plane seems to be the XY plane instead of the YZ plane (Ref Figure 2). Please check it.

10.  Line no. 329-320: “it can be found that the honeycomb core and panel of honeycomb sandwich panel”. Re-phase the sentence.

11.  Please look out for and correct grammatical errors.

Recommended with minor changes.

Author Response

We thank you for your appreciation of our contributions. We address your concerns in the revised version of this paper. We are very sorry for the mistakes in this manuscript and inconvenience they caused in your reading. The manuscript has been thoroughly revised and edited by a native speaker. Detailed information of modification is listed below.

Reviewer 2 Report

The reviewed article deals with the design of broadband flexible microstrip antenna and antenna array using conformal load-bearing structure. The topics of the article are important, and the results shown are interesting. However, the perception of the article is negatively affected by the way the article is written. Here is a list of my criticisms (order does not imply validity):
- at the end of section 1, an identical paragraph is repeated twice
- formulas (3) describe a generally known qualitative correlation. However, it is impossible to calculate anything from them, despite what the authors claim in several places in the article
- the text is full of strange and unfinished sentences (e.g. 'the honeycomb layer is a periodic structure composed of the periodic cell structure, and the periodic side length is' - lines 161, 162). The article needs a thorough linguistic correction of the
- equation (7) - since the scattering parameters are dimensionless, the impedance Z is not expressed in ohms !!!
- the results shown in section 3.1. (The equivalent medium model of honeycomb core layer) have no effect on how the antenna described in the article is designed. Please show this relationship or remove this section
- An analogous comment applies to section 3.2. (Principle for wideband operation). Both sections currently give the impression of having been added only to justify references to the cited literature, which gives a very negative, arguably unfair impression of the authors' intentions
- it seems unlikely that the prototype shown in Figure 11 could be bent during measurements within 30-90 degrees, especially since there is an SMA connector rigidly soldered. If this was in fact the case, please post a relevant photo

Due to the numerous comments that impinge on the merits of the atic, I suggest a major revision.
Regards

Author Response

(The authors gave the same response as above.)

Reviewer 3 Report

The authors have presented a good work on paper titled “Broadband Flexible Microstrip Antenna Array with Conformal 2 Load-bearing Structure”. My suggestions are as follows:

1.       Mention the electrical length of the antenna and operating band in the abstract

2.       Bending is performed along one axis only, it should be performed along other axis as well.

3.       Compare the following flexible antennas with respect to size and operating bands:

a.       Dual Polarized, Multiband Four-Port Decagon Shaped Flexible MIMO Antenna for Next Generation Wireless Applications

b.       ‘A compact tri-band flexible MIMO antenna based on liquid crystal polymer for wearable applications

c.       Wideband Ring-Monopole Flexible Antenna with Stub for WLAN/C-Band/X-Band Applications

d.       Flexible Interconnected 4-Port MIMO Antenna for Sub-6 GHz 5G and X Band Applications

Author Response

(The authors gave the same response as above.)

Round 2

Reviewer 2 Report

I thank the authors for the explanations provided and for incorporating some of my comments into the text of the article. However, two points still need to be clarified:
1. "According to the transmission line theory, Z is the characteristic impedance which may refer to the literature titled “Measurement of the Intrinsic Properties of Materials by Time-Domain Techniques” (doi: 10.1109/tim.1970.4313932). In this paper, Z is not expressed in ohm."

It is clear from the text of the article indicated that Z (as characteristic impedance Z0) is in ohms, so I reiterate my question about formulas (7)-(10). Please carefully check the transformations performed. If the indicated article is so important why is it not cited instead of item [21]?

2 I share the authors' opinion that "To reduce the computer memory and computation time of CLA, the honeycomb core layer can be characterized as a homogeneous plane by the equivalent EM parameters." In the context of the reviewed article, it makes sense if the equivalent continous medium will be used in the CST simulation to simplify the model and reduce computation time. Was this done? If so - please write it clearly in the text of the article. If not - please do such antenna simulations in Section 4.

I suggest major revision again.
Regards

Author Response

Dear Editor and Reviewers,

Thanks very much for taking your time and effort to review this manuscript. We really appreciate all your comments and suggestions. Those are all valuable and very helpful for revising and improving our paper, as well as the important guiding significance to our researches. We have studied comments carefully and have made correction which we hope meet with approval. Revised portion are marked up using the “Track Changes” function in the paper. The main corrections in the paper and responds to the reviewer’s comments are in bottom of this mail.

We sincerely appreciate your consideration of our manuscript. If you have any questions, please don’t hesitate to contact me at the address below.

Yours Sincerely

Prof. Hongxing Zheng

Hebei University of Technology,

Xiping Road No. 5340,

Beichen District, Tianjin, China.

E-mail: hxzheng@hebut.edu.cn

Round 3

Reviewer 2 Report

No additional comments.